# SINGLE EPISODE POLICY TRANSFER IN REINFORCEMENT LEARNING

**Jiachen Yang**
Georgia Institute of Technology, USA
`jiachen.yang@gatech.edu`

**Brenden Petersen**
Lawrence Livermore National Laboratory, USA
`petersen33@llnl.gov`

**Hongyuan Zha**
Georgia Institute of Technology, USA
`zha@cc.gatech.edu`

**Daniel Faissol**
Lawrence Livermore National Laboratory, USA
`faissol1@llnl.gov`

## ABSTRACT

Transfer and adaptation to new unknown environmental dynamics is a key challenge for reinforcement learning (RL). An even greater challenge is performing near-optimally in a single attempt at test time, possibly without access to dense rewards, which is not addressed by current methods that require multiple experience rollouts for adaptation. To achieve single episode transfer in a family of environments with related dynamics, we propose a general algorithm that optimizes a probe and an inference model to rapidly estimate underlying latent variables of test dynamics, which are then immediately used as input to a universal control policy. This modular approach enables integration of state-of-the-art algorithms for variational inference or RL. Moreover, our approach does not require access to rewards at test time, allowing it to perform in settings where existing adaptive approaches cannot. In diverse experimental domains with a single episode test constraint, our method significantly outperforms existing adaptive approaches and shows favorable performance against baselines for robust transfer.

## 1 INTRODUCTION

One salient feature of human intelligence is the ability to perform well in a single attempt at a new task instance, by recognizing critical characteristics of the instance and immediately executing appropriate behavior based on experience in similar instances. Artificial agents must do likewise in applications where success must be achieved in one attempt and failure is irreversible. This problem setting, *single episode transfer*, imposes a challenging constraint in which an agent experiences—and is evaluated on—only *one* episode of a test instance.

As a motivating example, a key challenge in precision medicine is the uniqueness of each patient's response to therapeutics (Hodson, 2016; Bordbar *et al.*, 2015; Whirl-Carrillo *et al.*, 2012). Adaptive therapy is a promising approach that formulates a treatment strategy as a sequential decision-making problem (Zhang *et al.*, 2017; West *et al.*, 2018; Petersen *et al.*, 2019). However, heterogeneity among instances may require explicitly accounting for factors that underlie individual patient dynamics. For example, in the case of adaptive therapy for sepsis (Petersen *et al.*, 2019), predicting patient response prior to treatment is not possible. However, differences in patient responses can be observed via blood measurements very early after the onset of treatment (Cockrell and An, 2018).

As a first step to address *single episode transfer* in reinforcement learning (RL), we propose a general algorithm for near-optimal test-time performance in a family of environments where differences in dynamics can be ascertained early during an episode. Our key idea is to train an inference model and a probe that together achieve rapid inference of latent variables—which account for variation in a family of similar dynamical systems—using a small fraction (e.g., 5%) of the test episode, then deploy a universal policy conditioned on the estimated parameters for near-optimal control on the new instance. Our approach combines the advantages of robust transfer and adaptation-based transfer, as we learn a single universal policy that requires no further training during test, but which is adapted to the new environment by conditioning on an unsupervised estimation of new latent dynamics.

In contrast to methods that quickly adapt or train policies via gradients during test but assume access to multiple test rollouts and/or dense rewards (Finn *et al.*, 2017; Killian *et al.*, 2017; Rakelly *et al.*, 2019), we explicitly optimize for performance in one test episode without accessing the reward function at test time. Hence our method applies to real-world settings in which rewards during test are highly delayed or even completely inaccessible—e.g., a reward that depends on physiological factors that are accessible only in simulation and not from real patients. We also consider computation time a crucial factor for real-time application, whereas some existing approaches require considerable computation during test (Killian *et al.*, 2017). Our algorithm builds on variational inference and RL as submodules, which ensures practical compatibility with existing RL workflows.

Our main contribution is a simple general algorithm for single episode transfer in families of environments with varying dynamics, via rapid inference of latent variables and immediate execution of a universal policy. Our method attains significantly higher cumulative rewards, with orders of magnitude faster computation time during test, than the state-of-the-art model-based method (Killian *et al.*, 2017), on benchmark high-dimensional domains whose dynamics are discontinuous and continuous in latent parameters. We also show superior performance over optimization-based meta-learning and favorable performance versus baselines for robust transfer.

## 2 SINGLE EPISODE TRANSFER IN RL: PROBLEM SETUP

Our goal is to train a model that performs close to optimal within a single episode of a test instance with new unknown dynamics. We formalize the problem as a family $(\mathcal{S}, \mathcal{A}, \mathcal{T}, R, \gamma)$, where $(\mathcal{S}, \mathcal{A}, R, \gamma)$ are the state space, action space, reward function, and discount of an episodic Markov decision process (MDP). Each *instance* of the family is a stationary MDP with transition function $\mathcal{T}_z(s'|s, a) \in \mathcal{T}$. When a set $\mathcal{Z}$ of physical parameters determines transition dynamics (Konidaris and Doshi-Velez, 2014), each $\mathcal{T}_z$ has a hidden parameter $z \in \mathcal{Z}$ that is sampled once from a distribution $P_{\mathcal{Z}}$ and held constant for that instance. For more general stochastic systems whose modes of behavior are not easily attributed to physical parameters, $\mathcal{Z}$ is induced by a generative latent variable model that indirectly associates each $\mathcal{T}_z$ to a latent variable $z$ learned from observed trajectory data. We refer to "latent variable" for both cases, with the clear ontological difference understood. Depending on application, $\mathcal{T}_z$ can be continuous or discontinuous in $z$. We strictly enforce the challenging constraint that latent variables are never observed, in contrast to methods that use known values during training (Yu *et al.*, 2017), to ensure the framework applies to challenging cases without prior knowledge.

This formulation captures a diverse set of important problems. Latent space $\mathcal{Z}$ has physical meaning in systems where $\mathcal{T}_z$ is a continuous function of physical parameters (e.g., friction and stiffness) with unknown values. In contrast, a discrete set $\mathcal{Z}$ can induce qualitatively different dynamics, such as a 2D navigation task where $z \in \{0, 1\}$ decides if the same action moves in either a cardinal direction or its opposite (Killian *et al.*, 2017). Such drastic impact of latent variables may arise when a single drug is effective for some patients but causes serious side effects for others (Cockrell and An, 2018).

**Training phase.** Our training approach is fully compatible with RL for episodic environments. We sample many instances, either via a simulator with controllable change of instances or using off-policy batch data in which demarcation of instances—but not values of latent variables—is known, and train for one or more episodes on each instance. While we focus on the case with known change of instances, the rare case of unknown demarcation can be approached either by preprocessing steps such as clustering trajectory data or using a dynamic variant of our algorithm (Appendix C).

**Single test episode.** In contrast to prior work that depend on the luxury of multiple experience rollouts for adaptation during test time (Doshi-Velez and Konidaris, 2016; Killian *et al.*, 2017; Finn *et al.*, 2017; Rakelly *et al.*, 2019), we introduce the strict constraint that the trained model has access to—and is evaluated on—*only one* episode of a new test instance. This reflects the need to perform near-optimally as soon as possible in critical applications such as precision medicine, where an episode for a new patient with new physiological dynamics is the entirety of hospitalization.

## 3 SINGLE EPISODE POLICY TRANSFER

We present Single Episode Policy Transfer (SEPT), a high-level algorithm for single episode transfer between MDPs with different dynamics. The following sections discuss specific design choices in SEPT, all of which are combined in synergy for near-optimal performance in a single test episode.

## 3.1 POLICY TRANSFER THROUGH LATENT SPACE

Our best theories of natural and engineered systems involve physical constants and design parameters that enter into dynamical models. This physicalist viewpoint motivates a partition for transfer learning in families of MDPs: 1. learn a representation of latent variables with an inference model that rapidly encodes a vector $\hat{z}$ of discriminative features for a new instance; 2. train a universal policy $\pi(a|s, z)$ to perform near-optimally for dynamics corresponding to any latent variable in $\mathcal{Z}$; 3. immediately deploy both the inference model and universal policy on a given test episode. To build on the generality of model-free RL, and for scalability to systems with complex dynamics, we do not expend computational effort to learn a model of $\mathcal{T}_z(s'|s, a)$, in contrast to model-based approaches (Killian *et al.*, 2017; Yao *et al.*, 2018). Instead, we leverage expressive variational inference models to represent latent variables and provide uncertainty quantification.

In domains with ground truth hidden parameters, a latent variable encoding is the most succinct representation of differences in dynamics between instances. As the encoding $\hat{z}$ is held constant for all episodes of an instance, a universal policy $\pi(a|s, z)$ can either adapt to all instances when $\mathcal{Z}$ is finite, or interpolate between instances when $\mathcal{T}_z$ is continuous in $z$ (Schaul *et al.*, 2015). Estimating a discriminative encoding for a new instance enables immediate deployment of $\pi(a|s, z)$ on the single test episode, bypassing the need for further fine-tuning. This is critical for applications where further training complex models on a test instance is not permitted due to safety concerns. In contrast, methods that do not explicitly estimate a latent representation of varied dynamics must use precious experiences in the test episode to tune the trained policy (Finn *et al.*, 2017).

In the training phase, we generate an optimized[1] dataset $\mathcal{D} := \{\tau^i\}_{i=1}^N$ of short trajectories, where each $\tau^i := (s_1^i, a_1^i, \ldots, s_{T_p}^i, a_{T_p}^i)$ is a sequence of early state-action pairs at the start of episodes of instance $\mathcal{T}_i \in \mathcal{T}$ (e.g. $T_p = 5$). We train a variational auto-encoder, comprising an approximate posterior inference model $q_\phi(z|\tau)$ that produces a latent encoding $\hat{z}$ from $\tau$ and a parameterized generative model $p_\psi(\tau|z)$. The dimension chosen for $\hat{z}$ may differ from the exact true dimension when it exists but is unknown; domain knowledge can aid the choice of dimensionality reduction. Because dynamics of a large variety of natural systems are determined by independent parameters (e.g., coefficient of contact friction and Reynolds number can vary independently), we consider a disentangled latent representation where latent units capture the effects of independent generative parameters. To this end, we bring $\beta$-VAE (Higgins *et al.*, 2017) into the context of families of dynamical systems, choosing an isotropic unit Gaussian as the prior and imposing the constraint $D_{KL}(q_\phi(z|\tau^i)\|p(z)) < \epsilon$. The $\beta$-VAE is trained by maximizing the variational lower bound $\mathcal{L}(\psi, \phi; \tau^i)$ for each $\tau^i$ across $\mathcal{D}$:

$$\max_{\psi,\phi} \log p_\psi(\tau^i) \geq \mathcal{L}(\psi, \phi; \tau^i) := -\beta D_{KL}(q_\phi(z|\tau^i)\|p(z)) + \mathbb{E}_{q_\phi(z|\tau^i)}\big[\log p_\psi(\tau^i|z)\big] \quad (1)$$

This subsumes the VAE (Kingma and Welling, 2014) as a special case ($\beta = 1$), and we refer to both as VAE in the following. Since latent variables only serve to differentiate among trajectories that arise from different transition functions, the meaning of latent variables is not affected by isometries and hence the value of $\hat{z}$ by itself need not have any simple relation to a physically meaningful $z$ even when one exists. Only the partition of latent space is important for training a universal policy.

Earlier methods for a family of similar dynamics relied on Bayesian neural network (BNN) approximations of the entire transition function $s_{t+1} \sim \hat{\mathcal{T}}_z^{(\text{BNN})}(s_t, a_t)$, which was either used to perform computationally expensive fictional rollouts during test time (Killian *et al.*, 2017) or used indirectly to further optimize a posterior over $z$ (Yao *et al.*, 2018). Our use of variational inference is more economical: the encoder $q_\phi(z|\tau)$ can be used immediately to infer latent variables during test, while the decoder $p_\psi(\tau|z)$ plays a crucial role for optimized probing in our algorithm (see Section 3.3).

In systems with ground truth hidden parameters, we desire two additional properties. The encoder should produce low-variance encodings, which we implement by minimizing the entropy of $q_\phi(z|\tau)$:

$$\min_\phi H(q_\phi(z|\tau)) := -\int_z q_\phi(z|\tau) \log q_\phi(z|\tau) dz = \frac{D}{2}\log(2\pi) + \frac{1}{2}\sum_{d=1}^D \big(1 + \log \sigma_d^2\big) \quad (2)$$

under a diagonal Gaussian parameterization, where $\sigma_d^2 = \text{Var}(q_\phi(z|\tau))$ and $\dim(z) = D$. We add $-H(q_\phi(z|\tau))$ as a regularizer to equation 1. Second, we must capture the impact of $z$ on higher-

---

[1]In the sense of machine teaching, as explained fully in Section 3.3

order dynamics. While previous work neglects the order of transitions $(s_t, a_t, s_{t+1})$ in a trajectory (Rakelly *et al.*, 2019), we note that a single transition may be compatible with multiple instances whose differences manifest only at higher orders. In general, partitioning the latent space requires taking the ordering of a temporally-extended trajectory into account. Therefore, we parameterize our encoder $q_\phi(z|\tau)$ using a bidirectional LSTM—as both temporal directions of $(s_t, a_t)$ pairs are informative—and we use an LSTM decoder $p_\psi(\tau|z)$ (architecture in Appendix E.2). In contrast to embedding trajectories from a *single* MDP for hierarchical learning (Co-Reyes *et al.*, 2018), our purpose is to encode trajectories from *different instances* of transition dynamics for optimal control.

## 3.2 TRANSFER OF A UNIVERSAL POLICY

We train a single universal policy $\pi(a|s, z)$ and deploy the same policy during test (without further optimization), for two reasons: robustness against imperfection in latent variable representation and significant improvement in scalability. Earlier methods trained multiple optimal policies $\{\pi_i^*(a|s)\}_{i=1}^N$ on training instances with a set $\{z^i\}_{i=1}^N$ of hidden parameters, then employed either behavioral cloning (Yao *et al.*, 2018) or off-policy Q-learning (Arnekvist *et al.*, 2019) to train a final policy $\pi(a|s, z)$ using a dataset $\{(s_t, \hat{z}^i; a_t \sim \pi_i^*(a|s_t))\}$. However, this supervised training scheme may not be robust (Yu *et al.*, 2017): if $\pi(a|s, z)$ were trained only using instance-specific optimal state-action pairs generated by $\pi_i^*(a|s)$ and posterior samples of $\hat{z}$ from an optimal inference model, it may not generalize well when faced with states and encodings that were not present during training. Moreover, it is computationally infeasible to train a collection $\{\pi_i^*\}_{i=1}^N$—which is thrown away during test—when faced with a large set of training instances from a continuous set $\mathcal{Z}$. Instead, we interleave training of the VAE and a single policy $\pi(a|s, z)$, benefiting from considerable computation savings at training time, and higher robustness due to larger effective sample count.

## 3.3 OPTIMIZED PROBING FOR ACCELERATED LATENT VARIABLE INFERENCE

To execute near-optimal control within a single test episode, we first rapidly compute $\hat{z}$ using a short trajectory of initial experience. This is loosely analogous to the use of preliminary medical treatment to define subsequent prescriptions that better match a patient's unique physiological response. Our goal of rapid inference motivates two algorithmic design choices to optimize this initial phase. First, the trajectory $\tau$ used for inference by $q_\phi(z|\tau)$ must be optimized, in the sense of machine teaching (Zhu *et al.*, 2018), as certain trajectories are more suitable than others for inferring latent variables that underlie system dynamics. If specific degrees of freedom are impacted the most by latent variables, an agent should probe exactly those dimensions to produce an informative trajectory for inference. Conversely, methods that deploy a single universal policy without an initial probing phase (Yao *et al.*, 2018) can fail in adversarial cases, such as when the initial placeholder $\hat{z}$ used in $\pi_\theta(a|s, \cdot)$ at the start of an instance causes failure to exercise dimensions of dynamics that are necessary for inference. Second, the VAE must be specifically trained on a dataset $\mathcal{D}$ of short trajectories consisting of initial steps of each training episode. We cannot expend a long trajectory for input to the encoder during test, to ensure enough remaining steps for control. Hence, single episode transfer motivates the machine teaching problem of learning to distinguish among dynamics: our algorithm must have learned both to generate and to use a short initial trajectory to estimate a representation of dynamics for control.

Our key idea of optimized probing for accelerated latent variable inference is to train a dedicated probe policy $\pi_\varphi(a|s)$ to generate a dataset $\mathcal{D}$ of short trajectories at the beginning of all training episodes, such that the VAE's performance on $\mathcal{D}$ is optimized[2]. Orthogonal to training a meta-policy for faster exploration *during* standard RL training (Xu *et al.*, 2018), our probe and VAE are trained for the purpose of performing well on a *new* test MDP. For ease of exposition, we discuss the case with access to a simulator, but our method easily allows use of off-policy batch data. We start each training episode using $\pi_\varphi$ for a *probe phase* lasting $T_p$ steps, record the probe trajectory $\tau_p$ into $\mathcal{D}$, train the VAE using minibatches from $\mathcal{D}$, then use $\tau_p$ with the encoder to generate $\hat{z}$ for use by $\pi_\theta(a|s, \hat{z})$ to complete the remainder of the episode (Algorithm 1). At test time, SEPT only requires lines 5, 8, and 9 in Algorithm 1 (training step in 9 removed; see Algorithm 2). The reward function for $\pi_\varphi$ is defined as the VAE objective, approximated by the variational lower bound (1): $R_p(\tau) := \mathcal{L}(\psi, \phi; \tau) \leq \log p_\psi(\tau)$. This feedback loop between the probe and VAE directly trains the

---

[2]In general, $\mathcal{D}$ is not related to the replay buffer commonly used in off-policy RL algorithms.

---

**Algorithm 1** Single Episode Policy Transfer: training phase

1: **procedure** SEPT-TRAIN
2:     Initialize encoder $\phi$, decoder $\psi$, probe policy $\varphi$, control policy $\theta$, and trajectory buffer $\mathcal{D}$
3:     **for** each instance $\mathcal{T}_z$ with transition function sampled from $\mathcal{T}$ **do**
4:         **for** each episode on instance $\mathcal{T}_z$ **do**
5:             Execute $\pi_\varphi$ for $T_p$ steps and store trajectory $\tau_p$ into $\mathcal{D}$
6:             Use variational lower bound (1) as the reward to train $\pi_\varphi$ by descending gradient (3)
7:             Train VAE using minibatches from $\mathcal{D}$ for gradient ascent on (1) and descent on (2)
8:             Estimate $\hat{z}$ from $\tau_p$ using encoder $q_\phi(z|\tau)$
9:             Execute $\pi_\theta(a|s,z)$ with $\hat{z}$ for remaining time steps and train it with suitable RL algorithm
10:         **end for**
11:     **end for**
12: **end procedure**

---

probe to help the VAE's inference of latent variables that distinguish different dynamics (Figure 1). We provide detailed justification as follows. First we state a result derived in Appendix A:

**Proposition 1.** *Let $p_\varphi(\tau)$ denote the distribution of trajectories induced by $\pi_\varphi$. Then the gradient of the entropy $H(p_\varphi(\tau))$ is given by*

$$\nabla_\varphi H(p_\varphi(\tau)) = \mathbb{E}_{p_\varphi(\tau)}\Big[\nabla_\varphi \sum_{i=0}^{T_p-1} \log(\pi_\varphi(a_i|s_i))(-\log p_\varphi(\tau))\Big] \tag{3}$$

Noting that dataset $\mathcal{D}$ follows distribution $p_\varphi$ and that the VAE is exactly trained to maximize the log probability of $\mathcal{D}$, we use $\mathcal{L}(\psi,\phi;\tau)$ as a tractable lowerbound on $\log p_\varphi(\tau)$. Crucially, to generate optimal probe trajectories for the VAE, we take a minimum-entropy viewpoint and *descend* the gradient (3). This is opposite of a maximum entropy viewpoint that encourages the policy to generate diverse trajectories (Co-Reyes *et al.*, 2018), which would minimize $\log p_\varphi(\tau)$ and produce an adversarial dataset for the VAE—hence, optimal probing is not equivalent to diverse exploration. The degenerate case of $\pi_\varphi$ learning to "stay still" for minimum entropy is precluded by any source of environmental stochasticity: trajectories from different instances will still differ, so degenerate trajectories result in low VAE performance. Finally we observe that equation 3 is the defining equation of a simple policy gradient algorithm (Williams, 1992) for training $\pi_\varphi$, with $\log p_\varphi(\tau)$ interpreted as the cumulative reward of a trajectory generated by $\pi_\varphi$. This completes our justification for defining reward $R_p(\tau) := \mathcal{L}(\psi,\phi;\tau)$. We also show empirically in ablation experiments that this reward is more effective than choices that encourage high perturbation of state dimensions or high entropy (Section 6).

The VAE objective function may not perfectly evaluate a probe trajectory generated by $\pi_\varphi$ because the objective value increases due to VAE training regardless of $\pi_\varphi$. To give a more stable reward signal to $\pi_\varphi$, we can use a second VAE whose parameters slowly track the main VAE according to $\psi' \leftarrow \alpha\psi + (1-\alpha)\psi'$ for $\alpha \in [0,1]$, and similarly for $\phi'$. While analogous to target networks in DQN (Mnih *et al.*, 2015), the difference is that our second VAE is used to compute the *reward* for $\pi_\varphi$.

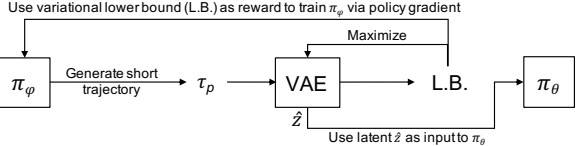

Figure 1: $\pi_\varphi$ learns to generate an optimal dataset for the VAE, whose performance is the reward for $\pi_\varphi$. Encoding $\hat{z}$ by the VAE is given to control policy $\pi_\theta$.

## 4 RELATED WORK

Transfer learning in a family of MDPs with different dynamics manifests in various formulations (Taylor and Stone, 2009). Analysis of $\epsilon$-stationary MDPs and $\epsilon$-MDPs provide theoretical grounding by showing that an RL algorithm that learns an optimal policy in an MDP can also learn a near-optimal policy for multiple transition functions (Kalmár *et al.*, 1998; Szita *et al.*, 2002).

Imposing more structure, the hidden-parameter Markov decision process (HiP-MDP) formalism posits a space of hidden parameters that determine transition dynamics, and implements transfer by model-based policy training after inference of latent parameters (Doshi-Velez and Konidaris, 2016; Konidaris and Doshi-Velez, 2014). Our work considers HiP-MDP as a widely applicable yet special case of a general viewpoint, in which the existence of hidden parameters is not assumed but rather is induced by a latent variable inference model. The key structural difference from POMDPs (Kaelbling *et al.*, 1998) is that given fixed latent values, each instance from the family is an MDP with no hidden states; hence, unlike in POMDPs, tracking a history of observations provides no benefit. In contrast to multi-task learning (Caruana, 1997), which uses the same tasks for training and test, and in contrast to parameterized-skill learning (Da Silva *et al.*, 2012), where an agent learns from a collection of rewards with given task identities in one environment with fixed dynamics, our training and test MDPs have different dynamics and identities of instances are not given.

Prior latent variable based methods for transfer in RL depend on a multitude of optimal policies during training (Arnekvist *et al.*, 2019), or learn a surrogate transition model for model predictive control with real-time posterior updates during test (Perez *et al.*, 2018). Our variational model-free approach does not incur either of these high computational costs. We encode trajectories to infer latent representation of differing dynamics, in contrast to state encodings in (Zhang *et al.*, 2018). Rather than formulating variational inference in the space of optimal value functions (Tirinzoni *et al.*, 2018), we implement transfer through variational inference in a latent space that underlies dynamics. Previous work for transfer across dynamics with hidden parameters employ model-based RL with Gaussian process and Bayesian neural network (BNN) models of the transition function (Doshi-Velez and Konidaris, 2016; Killian *et al.*, 2017), which require computationally expensive fictional rollouts to train a policy from scratch during test time and poses difficulties for real-time test deployment. DPT uses a fully-trained BNN to further optimize latent variable during a single test episode, but faces scalability issues as it needs one optimal policy per training instance (Yao *et al.*, 2018). In contrast, our method does not need a transition function and can be deployed without optimization during test. Methods for robust transfer either require access to multiple rounds from the test MDP during training (Rajeswaran *et al.*, 2017), or require the distribution over hidden variables to be known or controllable (Paul *et al.*, 2019). While meta-learning (Finn *et al.*, 2017; Rusu *et al.*, 2019; Zintgraf *et al.*, 2019; Rakelly *et al.*, 2019) in principle can take one gradient step during a single test episode, prior empirical evaluation were not made with this constraint enforced, and adaptation during test is impossible in settings without dense rewards.

## 5 Experimental Setup

We conducted experiments on three benchmark domains with diverse challenges to evaluate the performance, speed of reward attainment, and computational time of SEPT versus five baselines in the single test episode[3]. We evaluated four ablation and variants of SEPT to investigate the necessity of all algorithmic design choices. For each method on each domain, we conducted 20 independent training runs. For each trained model, we evaluate on $M$ independent test instances, all starting with the same model; adaptations during the single test episode, if done by any method, are not preserved across the independent test instances. This means we evaluate on a total of $20M$ independent test instances per method per domain. Hyperparameters were adjusted using a coarse coordinate search on validation performance. We used DDQN with prioritized replay (Van Hasselt *et al.*, 2016; Schaul *et al.*, 2016) as the base RL component of all methods for a fair evaluation of transfer performance; other RL algorithms can be readily substituted.

**Domains.** We use the same continuous state discrete action HiP-MDPs proposed by Killian *et al.* (2017) for benchmarking. Each isolated instance from each domain is solvable by RL, but it is highly challenging, if not impossible, for naïve RL to perform optimally for all instances because significantly different dynamics require different optimal policies. In **2D navigation**, dynamics are discontinuous in $z \in \{0, 1\}$ as follows: location of barrier to goal region, flipped effect of actions (i.e., depending on $z$, the *same* action moves in either a cardinal direction or its opposite), and direction of a nonlinear wind. In **Acrobot** (Sutton *et al.*, 1998), the agent applies $\{+1, 0, -1\}$ torques to swing a two-link pendulum above a certain height. Dynamics are determined by a vector $z = (m_1, m_2, l_1, l_2)$ of masses and lengths, centered at 1.0. We use four unique instances in training and validation,

---

[3]Code for all experiments is available at `https://github.com/011235813/SEPT`.

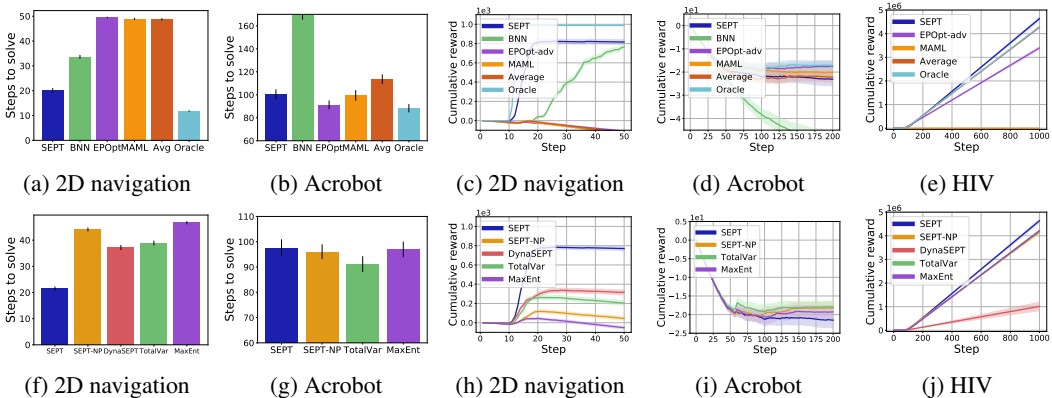

Figure 2: (a-e): Comparison against baselines. (a-b): Number of steps to solve 2D navigation and Acrobot in a single test episode; failure to solve is assigned a count of 50 in 2D nav and 200 in Acrobot. (c-e): Cumulative reward versus test episode step. BNN requires long computation time and showed low rewards on HIV, hence we report 3 seeds in Figure 4b. (f-j): Ablation results. DynaSEPT is out of range in (g), see Figure 4a. Error bars show standard error of mean over all test instances over 20 training runs per method.

constructed by sampling $\Delta z$ uniformly from $\{-0.3, -0.1, 0.1, 0.3\}$ and adding it to all components of $z$. During test, we sample $\Delta z$ from $\{-0.35, -0.2, 0.2, 0.35\}$ to evaluate both interpolation and extrapolation. In **HIV**, a patient's state dynamics is modeled by differential equations with high sensitivity to 12 hidden variables and separate steady-state regions of health, such that different patients require unique treatment policies (Adams *et al.*, 2004). Four actions determine binary activation of two drugs. We have $M = 10, 5, 5$ for 2D navigation, Acrobot, and HIV, respectively.

**Baselines.** First, we evaluated two simple baselines that establish approximate bounds on test performance of methods that train a single policy: as a lower bound, **Avg** trains a single policy $\pi(a|s)$ on all instances sampled during training and runs directly on test instances; as an upper bound in the limit of perfect function approximation for methods that use latent variables as input, **Oracle** $\pi(a|s, z)$ receives the true hidden parameter $z$ during both training and test. Next we adapted existing methods, detailed in Appendix E.1, to single episode test evaluation: 1. we allow the model-based method **BNN** (Killian *et al.*, 2017) to fine-tune a pre-trained BNN and train a policy using BNN-generated fictional episodes every 10 steps during the test episode; 2. we adapted the adversarial part of EPOpt (Rajeswaran *et al.*, 2017), which we term **EPOpt-adv**, by training a policy $\pi(a|s)$ on instances with the lowest 10-percentile performance; 3. we evaluate **MAML** as an archetype of meta-learning methods that require dense rewards or multiple rollouts (Finn *et al.*, 2017). We allow MAML to use a trajectory of the same length as SEPT's probe trajectory for one gradient step during test. We used the same architecture for the RL module of all methods (Appendix E.2). To our knowledge, these model-free baselines are evaluated on single-episode transfer for the first time in this work.

**Ablations.** To investigate the benefit of our optimized probing method for accelerated inference, we designed an ablation called **SEPT-NP**, in which trajectories generated by the control policy are used by the encoder for inference and stored into $\mathcal{D}$ to train the VAE. Second, we investigated an alternative reward function for the probe, labeled **TotalVar** and defined as $R(\tau) := 1/T_p \sum_{t=1}^{T_p-1} \sum_{i=1}^{\dim(\mathcal{S})} |s_{t+1,i} - s_{t,i}|$ for probe trajectory $\tau$. In contrast to the minimum entropy viewpoint in Section 3.3, this reward encourages generation of trajectories that maximize total variation across all state space dimensions. Third, we tested the maximum entropy viewpoint on probe trajectory generation, labeled **MaxEnt**, by giving *negative* lowerbound as the probe reward: $R_p(\tau) := -\mathcal{L}(\psi, \phi; \tau)$. Last, we tested whether **DynaSEPT**, an extension that dynamically decides to probe or execute control (Appendix C), has any benefit for stationary dynamics.

## 6 RESULTS AND DISCUSSION

2D navigation and Acrobot are solved upon attaining terminal reward of 1000 and 10, respectively. SEPT outperforms all baselines in 2D navigation and takes significantly fewer number of steps to solve (Figures 2a and 2c). While a single instance of 2D navigation is easy for RL, handling multiple

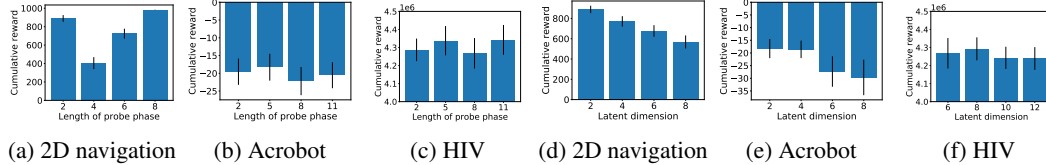

(a) 2D navigation    (b) Acrobot    (c) HIV    (d) 2D navigation    (e) Acrobot    (f) HIV

Figure 3: Cumulative reward on test episode for different $T_p$ (a-c) and different $\dim(z)$ (d-f). $8M$ independent test instances for each hyperparameter setting.

instances is highly non-trivial. EPOpt-adv and Avg almost never solve the test instance—we set "steps to solve" to 50 for test episodes that were unsolved—because interpolating between instance-specific optimal policies in policy parameter space is not meaningful for any task instance. MAML did not perform well despite having the advantage of being provided with rewards at test time, unlike SEPT. The gradient adaptation step was likely ineffective because the rewards are sparse and delayed. BNN requires significantly more steps than SEPT, and it uses four orders of magnitude longer computation time (Table 4), due to training a policy from scratch during the test episode. Training times of all algorithms except BNN are in the same order of magnitude (Table 3).

In Acrobot and HIV, where dynamics are continuous in latent variables, interpolation within policy space can produce meaningful policies, so all baselines are feasible in principle. SEPT is statistically significantly faster than BNN and Avg, is within error bars of MAML, while EPOpt-adv outperforms the rest by a small margin (Figures 2b and 2d). Figure 5 shows that SEPT is competitive in terms of percentage of solved instances. As the true values of latent variables for Acrobot test instances were interpolated and extrapolated from the training values, this shows that SEPT is robust to out-of-training dynamics. BNN requires more steps due to simultaneously learning and executing control during the test episode. On HIV, SEPT reaches significantly higher cumulative rewards than all methods. Oracle is within the margin of error of Avg. This may be due to insufficient examples of the high-dimensional ground truth hidden parameters. Due to its long computational time, we run three seeds for BNN on HIV, shown in Figure 4b, and find it was unable to adapt within one test episode.

Comparing directly to reported results in DPT (Yao *et al.*, 2018), SEPT solves 2D Navigation at least 33% (>10 steps) faster, and the cumulative reward of SEPT (mean and standard error) are above DPT's mean cumulative reward in Acrobot (Table 2). Together, these results show that methods that explicitly distinguish different dynamics (e.g., SEPT and BNN) can significantly outperform methods that implicitly interpolate in policy parameter space (e.g., Avg and EPOpt-adv) in settings where $z$ has large discontinuous effect on dynamics, such as 2D navigation. When dynamics are continuous in latent variables (e.g., Acrobot and HIV), interpolation-based methods fare better than BNN, which faces the difficulty of learning a model of the entire family of dynamics. SEPT worked the best in the first case and is robust to the second case because it explicitly distinguishes dynamics and does not require learning a full transition model. Moreover, SEPT does not require rewards at test time allowing it be useful on a broader class of problems than optimization-based meta-learning approaches like MAML. Appendix D contains training curves.

**Ablation results.** Comparing to SEPT-NP, Figures 2f, 2g and 2j show that the probe phase is necessary to solve 2D navigation quickly, while giving similar performance in Acrobot and significant improvement in HIV. SEPT significantly outperformed TotalVar in 2D navigation and HIV, while TotalVar gives slight improvement in Acrobot, showing that directly using VAE performance as the reward for probing in certain environments can be more effective than a reward that deliberately encourages perturbation of state dimensions. The clear advantage of SEPT over MaxEnt in 2D navigation and HIV supports our hypothesis in Section 3.3 that the variational lowerbound, rather than its negation in the maximum entropy viewpoint, should be used as the probe reward, while performance was not significantly differentiated in Acrobot. SEPT outperforms DynaSEPT on all problems where dynamics are stationary during each instance. On the other hand, DynaSEPT is the better choice in a non-stationary variant of 2D navigation where the dynamics "switch" abruptly at $t = 10$ (Figure 4c).

**Robustness.** Figure 3 shows that SEPT is robust to varying the probe length $T_p$ and $\dim(z)$. Even with certain suboptimal probe length and $\dim(z)$, it can outperform all baselines on 2D navigation in both steps-to-solve and final reward; it is within error bars of all baselines on Acrobot based on final

cumulative reward; and final cumulative reward exceeds that of baselines in HIV. Increasing $T_p$ means foregoing valuable steps of the control policy and increasing difficulty of trajectory reconstruction for the VAE in high dimensional state spaces; $T_p$ is a hyper-parameter that should be validated for each application. Appendix D.5 shows the effect of $\beta$ on latent variable encodings.

## 7 CONCLUSION AND FUTURE DIRECTIONS

We propose a general algorithm for single episode transfer among MDPs with different stationary dynamics, which is a challenging goal with real-world significance that deserves increased effort from the transfer learning and RL community. Our method, Single Episode Policy Transfer (SEPT), trains a probe policy and an inference model to discover a latent representation of dynamics using very few initial steps in a single test episode, such that a universal policy can execute optimal control without access to rewards at test time. Strong performance versus baselines in domains involving both continuous and discontinuous dependence of dynamics on latent variables show the promise of SEPT for problems where different dynamics can be distinguished via a short probing phase.

The dedicated probing phase may be improved by other objectives, in addition to performance of the inference model, to mitigate the risk and opportunity cost of probing. An open challenge is single episode transfer in domains where differences in dynamics of different instances are not detectable early during an episode, or where latent variables are fixed but dynamics are nonstationary. Further research on dynamic probing and control, as sketched in DynaSEPT, is one path toward addressing this challenge. Our work is one step along a broader avenue of research on general transfer learning in RL equipped with the realistic constraint of a single episode for adaptation and evaluation.

### ACKNOWLEDGMENTS

This work was performed under the auspices of the U.S. Department of Energy by Lawrence Livermore National Laboratory under contract DE-AC52-07NA27344. Lawrence Livermore National Security, LLC. LLNL-JRNL-791194.

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

## A  DERIVATIONS

**Proposition 1.** *Let $p_\varphi(\tau)$ denote the distribution of trajectories induced by $\pi_\varphi$. Then the gradient of the entropy $H(p_\varphi(\tau))$ is given by*

$$\nabla_\varphi H(p_\varphi(\tau)) = \mathbb{E}_{p_\varphi(\tau)}\Big[\nabla_\varphi \sum_{i=0}^{T_p-1} \log(\pi_\varphi(a_i|s_i))(-\log p_\varphi(\tau))\Big] \tag{3}$$

*Proof.* Assuming regularity, the gradient of the entropy is

$$\nabla_\varphi H(p_\varphi(\tau)) = -\nabla_\varphi \int p_\varphi(\tau) \log p_\varphi(\tau) d\tau$$

$$= -\int \nabla_\varphi p_\varphi(\tau) d\tau - \int \big(\nabla_\varphi p_\varphi(\tau)\big) \log p_\varphi(\tau) d\tau$$

$$= -\nabla_\varphi \int p_\varphi(\tau) d\tau - \int p_\varphi(\tau)\big(\nabla_\varphi \log p_\varphi(\tau)\big) \log p_\varphi(\tau) d\tau$$

$$= \mathbb{E}_{p_\varphi(\tau)}\big[\big(\nabla_\varphi \log p_\varphi(\tau)\big)\big(-\log p_\varphi(\tau)\big)\big]$$

For trajectory $\tau := (s_0, a_0, s_1, \ldots, s_t)$ generated by the probe policy $\pi_\varphi$:

$$p_\varphi(\tau) = p(s_0) \prod_{i=0}^{t-1} p(s_{i+1}|s_i, a_i)\pi_\varphi(a_i|s_i)$$

Then

$$\nabla_\varphi \log p_\varphi(\tau) = \nabla_\varphi\Big(\log p(s_0) + \sum_{i=0}^{t-1} \log p(s_{i+1}|s_i, a_i) + \sum_{i=0}^{t-1} \log \pi_\varphi(a_i|s_i)\Big)$$

Since $p(s_0)$ and $p(s_{i+1}|s_i, a_i)$ do not depend on $\varphi$, we get

$$\nabla_\varphi \log p_\varphi(\tau) = \nabla_\varphi \sum_{i=0}^{t-1} \log \pi_\varphi(a_i|s_i)$$

Substituting this into the gradient of the entropy gives equation 3. $\qquad\square$

## B  TESTING PHASE OF SEPT

---
**Algorithm 2** Single Episode Policy Transfer: testing phase

---
1: **procedure** SEPT-TEST
2:     Restore trained decoder $\psi$, encoder $\phi$, probe policy $\varphi$, and control policy $\theta$
3:     Run probe policy $\pi_\varphi$ for $T_p$ time steps and record trajectory $\tau_p$
4:     Use $\tau_p$ with decoder $q_\phi(z|\tau)$ to estimate $\hat{z}$
5:     Use $\hat{z}$ with control policy $\pi_\theta(a|s, z)$ for the remaining duration of the test episode
6: **end procedure**

---

## C  DYNASEPT

In our problem formulation, it is not necessary to compute $\hat{z}$ at every step of the test episode, as each instance is a stationary MDP and change of instances is known. However, removing the common assumption of stationarity leads to time-dependent transition functions $\mathcal{T}_z(s'|s, a)$, which introduces problematic cases. For example, a length $T_p$ probing phase would fail if $z$ leads to a switch in dynamics at time $t > T_p$, such as when poorly understood drug-drug interactions lead to abrupt changes in dynamics during co-medication therapies (Kastrin *et al.*, 2018). Here we describe an alternative general algorithm for non-stationary dynamics, which we call DynaSEPT. We train a

single policy $\pi_\theta(a|s,z,\eta)$ that dynamically decides whether to probe for better inference or act to maximize the MDP reward $R_{\text{env}}$, based on a scalar-valued function $\eta\colon \mathbb{R} \to [0,1]$ representing the degree of uncertainty in posterior inference, which is updated at every time step. The total reward is $R_{\text{tot}}(\tau) := \eta R_p(\tau) + (1-\eta)R_{\text{env}}(\tau_f)$, where $\tau$ is a short sliding-window trajectory of length $T_p$, and $\tau_f$ is the final state of $\tau$. The history-dependent term $R_p(\tau)$ is equivalent to a delayed reward given for executing a sequence of probe actions. Following the same reasoning for SEPT, one choice for $R_p(\tau)$ is $\mathcal{L}(\phi,\psi;\tau)$. Assuming the encoder outputs variance $\sigma_i^2$ of each latent dimension, one choice for $\eta$ is a normalized standard deviation over all dimensions of the latent variable, i.e. $\eta := \frac{1}{D}\sum_{i=1}^{D}\sigma_i(q_\phi)/\sigma_{i,\max}(q_\phi)$ , where $\sigma_{i,\max}$ is a running max of $\sigma_i$.

Despite its novelty, we consider DynaSEPT only for rare nonstationary dynamics and merely as a baseline in the predominant case of stationary dynamics, where SEPT is our primary contribution. DynaSEPT does not have any clear advantage over SEPT when each instance $\mathcal{T}_z$ is a stationary MDP. DynaSEPT requires $\eta$ to start at 1.0, representing complete lack of knowledge about latent variables, and it still requires the choice of hyperparameter $T_p$. Only after $T_p$ steps can it use the uncertainty of $q_\phi(z|\tau)$ to adapt $\eta$ and continue to generate the sliding window trajectory to improve $\hat{z}$. By this time, SEPT has already generated an optimized sequence using $\pi_\varphi$ for the encoder to estimate $\hat{z}$. If a trajectory of length $T_p$ is sufficient for computing a good estimate of latent variables, then SEPT is expected to outperform DynaSEPT.

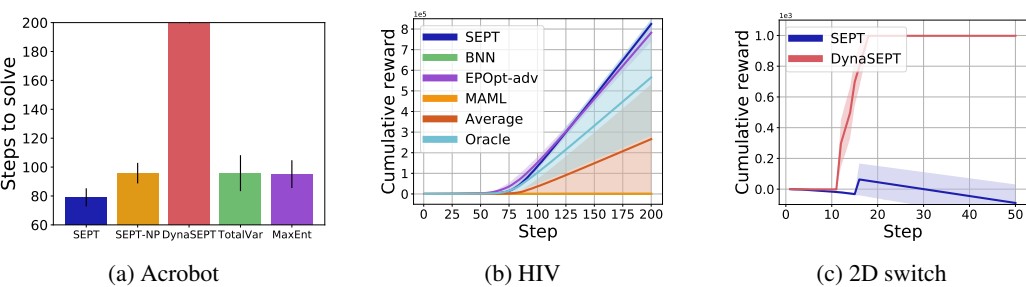

| (a) Acrobot | (b) HIV | (c) 2D switch |

Figure 4: (a) Ablations on Acrobot, including DynaSEPT with 3 seeds; (b) BNN and MAML attained orders of magnitude lower rewards than other baselines (3 seeds); (c) DynaSEPT performs well on nonstationary dynamics in 2D navigation.

# D  SUPPLEMENTARY EXPERIMENTAL RESULTS

## D.1  2D AND ACROBOT

2D navigation and Acrobot have a definition of "solved". Table 1 reports the number of steps in a test episode required to solve the MDP. Average and standard deviation were computed across all test instances and across all independently trained models. If an episode was not solved, the maximum allowed number of steps was used (50 for 2D navigation and 200 for Acrobot). Table 2 shows the mean and standard error of the cumulative reward over test episodes on Acrobot. The reported mean cumulative value for DPT in Acrobot is -27.7 (Yao *et al.*, 2018) .

Table 1: Steps to solve 2D navigation and Acrobot

| | 2D navigation | Acrobot |
|---|---|---|
| Average | 49±0.5 | 114±4.2 |
| Oracle | 12±0.3 | 88±3.6 |
| BNN | 34±0.8 | 169±4.0 |
| EPOpt-adv | 49±0.3 | 91±3.8 |
| MAML | 49±0.4 | 99±4.6 |
| SEPT | 20±0.9 | 100±4.4 |

Table 2: Error bars on Acrobot

| | Acrobot |
|---|---|
| Average | -22.2±2.3 |
| Oracle | -17.1±2.2 |
| BNN | -50.6±4.7 |
| EPOpt-adv | -17.5±2.2 |
| MAML | -20.1±2.6 |
| SEPT | -23.1±3.1 |

## D.2 TIMING COMPARISON

Table 3: Total training times in seconds on all experiment domains

|          | 2D navigation | Acrobot       | HIV         |
|----------|---------------|---------------|-------------|
| Average  | 1.3e3±277     | 1.0e3±85      | 1.4e3±47    |
| Oracle   | 0.6e3±163     | 1.1e3±129     | 1.5e3±47    |
| BNN      | 2.9e3±244     | 9.0e4±3.0e3   | 4.3e4±313   |
| EPOpt-adv| 1.1e3±44      | 1.1e3±1.0     | 1.9e3±33    |
| MAML     | 0.9e3±116     | 1.1e3±96      | 1.3e3±6.0   |
| SEPT     | 1.9e3±70      | 2.3e3±1e3     | 2.8e3±11    |

Table 4: Test episode time in seconds on all experiment domains

|          | 2D navigation | Acrobot       | HIV         |
|----------|---------------|---------------|-------------|
| Average  | 0.04±0.04     | 0.09±0.04     | 0.42±0.01   |
| Oracle   | 0.02±0.04     | 0.09±0.04     | 0.45±0.02   |
| BNN      | 2.6e3±957     | 2.8e3±968     | 1.4e3±8.8   |
| EPOpt-adv| 0.04±0.04     | 0.10±0.06     | 0.45±0.03   |
| MAML     | 0.05±0.05     | 0.10±0.07     | 0.48±0.01   |
| SEPT     | 0.04±0.07     | 0.12±0.10     | 0.60±0.02   |

## D.3 PERCENT OF SOLVED EPISODES

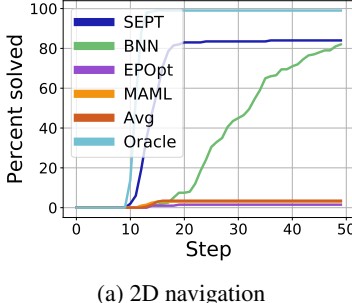

(a) 2D navigation

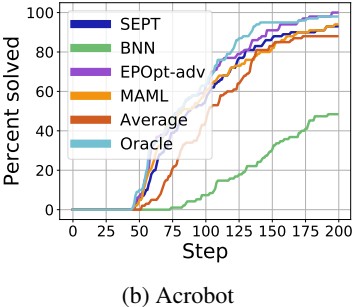

(b) Acrobot

Figure 5: Percent of solved test instances as a function of time steps during the test episode. Percentage is computed among 200 test instances for (a) 2D navigation and (b) 100 test instances for Acrobot.

2D navigation and Acrobot are considered solved upon reaching a terminal reward of 1000 and 10, respectively. Figure Figure 5 shows the percentage of all test episodes that are solved as a function of time steps in the test episode. Percentage is measured from a total of 200 test episodes for 2D navigation and 100 test episodes for Acrobot.

### D.4 TRAINING CURVES

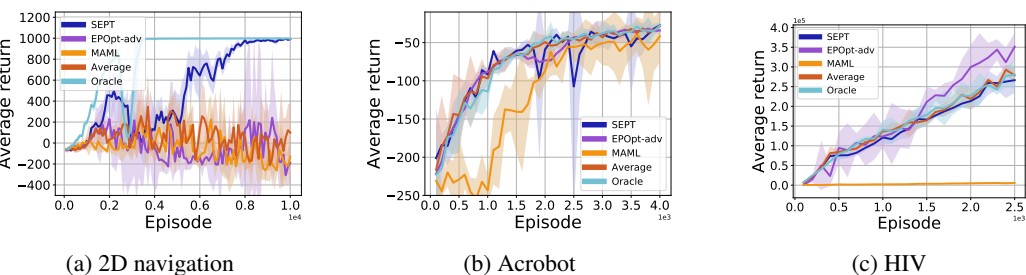

|(a) 2D navigation|(b) Acrobot|(c) HIV|

Figure 6: Average episodic return over training episodes. Only SEPT and Oracle converged in 2D navigation. All methods converged in Acrobot. All methods except MAML converged in HIV. BNN is not shown as the implementation (Killian *et al.*, 2017) does not record training progress.

Figure 6 shows training curves on all domains by all methods. None of the baselines, except for Oracle, converge in 2D navigation, because it is meaningless for Avg and EPOpt-adv to interpolate between optimal policies for each instance, and MAML cannot adapt due to lack of informative rewards for almost the entire test episode. Hence these baselines cannot work for a new unknown test episode, even in principle. We allowed the same number of training episodes for HIV as in Killian *et al.* (2017), and all baselines except MAML show learning progress.

### D.5 LATENT REPRESENTATION OF DYNAMICS

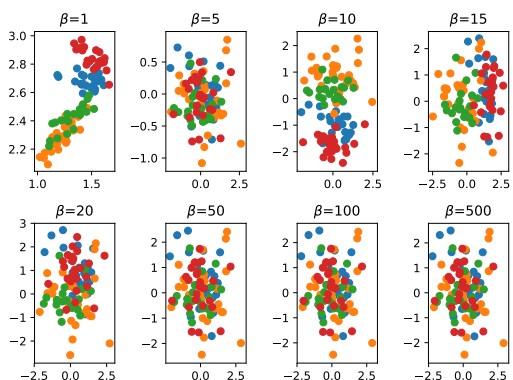

Figure 7: Two-dimensional encodings generated for four instances of Acrobot (represented by four ground-truth colors), for different values of $\beta$. We chose $\beta = 1$ for Acrobot.

There is a tradeoff between reconstruction and disentanglement as $\beta$ increases (Higgins *et al.*, 2017). Increasing $\beta$ encourages greater similarity between the posterior and an isotropic Gaussian. Figure 7 gives evidence that this comes at a cost of lower quality of separation in latent space.

## D.6 PROBE REWARD

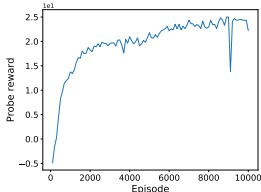

Figure 8: Probe policy reward curve in one training run in 2D navigation

## E EXPERIMENTAL DETAILS

For 2D navigation, Acrobot, and HIV, total number of training episodes allowed for all methods are 10k, 4k, and 2.5k, respectively. We switch instances once every 10, 8 and 5 episodes, respectively. There are 2, 8 and 5 unique training instances, and 2, 5, and 5 validation instances, respectively. For each independent training run, we tested on 10, 5, and 5 test instances, respectively.

### E.1 ALGORITHM IMPLEMENTATION DETAILS

The simple baselines Average and Oracle can be immediately deployed in a single test episode after training. However, the other methods for transfer learning require modification to work in the setting of single episode test, as they were not designed specifically for this highly constrained setting. We detail the necessary modifications below. We also describe the ablation SEPT-NP in more detail.

**BNN.** In Killian *et al.* (2017), a pre-trained BNN model was fine-tuned using the first test episode and then used to generate fictional episodes for training a policy from scratch. More episodes on the same test instance were allowed to help improve model accuracy of the BNN. In the single test episode setting, all fine-tuning and policy training must be conducted within the first test episode. We fine-tune the pre-trained BNN every 10 steps and allow the same total number of fictional episodes as reported in (Killian *et al.*, 2017) for policy training. We measured the cumulative reward attained by the policy—while it is undergoing training—during the single real test episode.

**EPOpt.** EPOpt trains on the lowest $\epsilon$-percentile rollouts from instances sampled from a source distribution, then adapts the source distribution using observations from the target instance (Rajeswaran *et al.*, 2017). Since we do not allow observation from the test instance, we only implemented the adversarial part of EPOpt. To run EPOpt with off-policy DDQN, we generated 100 rollouts per iteration and stored the lowest 10-percentile into the replay buffer, then executed the same number of minibatch training steps as the number that a regular DDQN would have done during rollouts.

**MAML.** While MAML uses many complete rollouts per gradient step (Finn *et al.*, 2017), the single episode test constraint mandates that it can only use a partial episode for adaptation during test, and hence the same must be done during meta-training. For both training and test, we allow MAML to take one gradient step for adaptation using a trajectory of the same length as the probe trajectory of SEPT, starting from the initial state of the episode. We implemented a first-order approximation that computes the meta-gradient at the post-update parameters but omits second derivatives. This was reported to have nearly equal performance as the full version, due to the use of ReLU activations.

**SEPT-NP.** $\pi_\theta(a|s,z)$ begins with a zero-vector for $z$ at the start of training. When it has produced a trajectory $\tau_p$ of length $T_p$, we store $\tau_p$ into $\mathcal{D}$ for training the VAE, and use $\tau_p$ with the VAE to estimate $z$ for the episode. Later training episodes begin with the rolling mean of all $z$ estimated so far. For test, we give the final rolling mean of $z$ at the end of training as initial input to $\pi_\theta(a|s,z)$.

### E.2 ARCHITECTURE

**Encoder.** For all experiments, the encoder $q_\phi(z|\tau)$ is a bidirectional LSTM with 300 hidden units and `tanh` activation. Outputs are mean-pooled over time, then fully-connected to two linear output layers of width dim($z$), interpreted as the mean and log-variance of a Gaussian over $z$.

**Decoder.** For all experiments, the decoder $p_\psi(\tau|z)$ is an LSTM with 256 hidden units and `tanh` activation. Given input $[s_t, a_t, \hat{z}]$ at LSTM time step $t$, the output is fully-connected to two linear output layers of width $|\mathcal{S}| + |\mathcal{A}|$, and interpreted as the mean and log-variance of a Gaussian decoder for the next state-action pair $(s_{t+1}, a_{t+1})$.

**Q network.** For all experiments, the $Q$ function is a fully-connected neural network with two hidden layers of width 256 and 512, ReLU activation, and a linear output layer of size $|\mathcal{A}|$. For SEPT and Oracle, the input is the concatenation $[s_t, z]$, where $z$ is estimated in the case of SEPT and $z$ is the ground truth in for the Oracle. For all other methods, the input is only the state $s$.

**Probe policy network.** For all experiments, $\pi_\varphi(a|s)$ is a fully-connected neural network with 3 hidden layers, ReLU activation, 32 nodes in all layers, and a softmax in the output layer.

### E.3   HYPERPARAMETERS

VAE learning rate was 1e-4 for all experiments. Size of the dataset $\mathcal{D}$ of probe trajectories was limited to 1000, with earliest trajectories discarded. 10 minibatches from $\mathcal{D}$ were used for each VAE training step. We used $\beta = 1$ for the VAE. Probe policy learning rate was 1e-3 for all experiments. DDQN minibatch size was 32, one training step was done for every 10 environment steps, $\epsilon_{\text{end}} = 0.15$, learning rate was 1e-3, gradient clip was 2.5, $\gamma = 0.99$, and target network update rate was 5e-3. Exploration decayed according $\epsilon_{n+1} = c\epsilon_n$ every episode, where $c$ satisfies $\epsilon_{\text{end}} = c^N \epsilon_{\text{start}}$ and $N$ is the total number of episodes. Prioritized replay used the same parameters in (Killian *et al.*, 2017).

Table 5: Hyperparameters used by each method, where applicable

|  | 2D navigation | Acrobot | HIV |
|---|---|---|---|
| $T_p$ | 2 | 5 | 8 |
| Instances | 1000 | 500 | 500 |
| Episodes per instance | 10 | 8 | 5 |
| VAE batch size | 10 | 64 | 64 |
| $\dim(\hat{z})$ | 2 | 2 | 6 |
| $\alpha$ | 1.0 | 0.005 | 1.0 |
| Probe minibatches | 1 | 10 | 1 |
| DDQN $\epsilon_{\text{start}}$ | 1.0 | 1.0 | 0.3 |

