# OpenReview forum: "Single Episode Policy Transfer in Reinforcement Learning"
_ICLR.cc/2020/Conference — Accept (Poster)_

### Official Review · AnonReviewer1 · 2019-10-22
**Official Blind Review #1**

**Rating:** 8

**Review:**

The main contribution of this paper is to learn a universal policy that is able to perform near-optimally on test tasks with transition dynamics that were never observed during training. This is achieved by using a "probe policy" to generate short trajectories that are then used to learn a latent encoding to categorise the transition dynamics of the current task. The universal policy is then conditions on both the state and this encoding so that the learned policy can perform well on tasks with different dynamics.

Overall I really like this paper. There has been a big push in the RL community to start evaluating algorithms on test tasks that are different to trained tasks and this paper takes a good step in this direction.

My main concern with this paper, which I would appreciate some feedback from the authors on, is regarding the trajectory length of the probe policy:

1. The method seems to rely heavily on the ability to learn an accurate latent encoding from a short trajectory which will may not be the case in many domains. It is easy to construct some domain where the dynamics behave identically initially but have some major differences deeper into the task at hand. I noted that the authors did mention this in their conclusion as an area for future research but I want to know if they have any high level ideas on how to rectify this because it seems that this will be a major limitation of the proposed algorithm in practice at least for certain domains.

2. In the experiments I can see that in some cases a longer probe performed more poorly than a shorter probe (for example Fig3a for 2D navigation. My intuition tells me that a longer probe should outperform a shorter probe since it contains at least as much information. Can the authors explain why this occurs?

**Experience Assessment:**

I have published one or two papers in this area.

**Review Assessment: Checking Correctness Of Derivations And Theory:**

I did not assess the derivations or theory.

**Review Assessment: Checking Correctness Of Experiments:**

I assessed the sensibility of the experiments.

**Review Assessment: Thoroughness In Paper Reading:**

I read the paper at least twice and used my best judgement in assessing the paper.

---

> ### Author Response · Authors · 2019-11-15
> **We thank Reviewer 1 for positive feedback on our work and on the importance of this research direction**
>
> We thank Reviewer 1 for positive feedback on both the importance of this research direction and our contributions in the paper.
>
> As Reviewer 1 correctly point out, SEPT is specifically designed for domains in which differences in dynamics can be ascertained early during an episode. There are two ways this can be violated. First, the dynamics itself may change during an episode, which violates the stationarity assumption of MDPs. We have preliminary ideas on how to account for this, which we explain in Appendix C of the paper: we proposed a dynamic variant of SEPT, called DynaSEPT, in which the policy dynamically decides to explore or execute control based on the uncertainty over the latent variable. We show in Figure 4d preliminary results that DynaSEPT can perform well when the dynamics of 2D navigation suddenly switches at an unknown time point late in the episode. Second, the dynamics can be stationary, but differences in dynamics of different instances are not detectable early during an episode. This is extremely challenging, even for human experts: if a patient is affected by a new and unknown strain of infection and the unique measurable symptoms of this strain are highly delayed, even experts face a significant challenge in delivering an optimal cure. An extension of DynaSEPT may be feasible here, by continually updating the latent variable for input to the control policy. Having said this, we believe that a large class of important physical systems do not pose these additional challenges and can be addressed by a method like SEPT. We hope that our contributions serve as a useful milestone for the RL community to push forward on the two challenging cases.
>
> While a longer probe trajectory contains as much information about dynamics as a shorter trajectory, it is also more difficult for the VAE to reconstruct the entire trajectory, especially for high dimensional state spaces, stochastic environments, and a limited capacity neural network for the VAE. As we have scaled up our experimental pipeline, we will verify the low performance of T_p = 4 in Figure 3a with more independent runs. The fact that T_p = 6 and 8 display equivalent performance in the relatively simple dynamics of 2D navigation supports the case that longer probe duration should not hurt performance, assuming enough capacity in the VAE. However, as shown in Figure 3(b-c), the optimal probe duration for different domains will be different, so in general a hyperparameter sweep over validation instances is needed.

---

### Official Review · AnonReviewer4 · 2019-10-22
**Official Blind Review #4**

**Rating:** 8

**Review:**

Summary

This paper addresses the problem of transfer in RL. After an agent is given an opportunity to train from a distribution of environments, we want an agent to perform well on the test environment. This paper specifically focuses on the setting where the state space, action space, reward space, and discount factor are the same across all environments, while the transition dynamics may differ. An environment's transition dynamics is assumed to depend on a hidden parameter that is not observed by the agent, in contrast to some previous work which assumes observability.

The idea of the main algorithm is as follows. During the training phase, a series of environments is presented. The agent is aware of the demarcations between the environments, but not of the identity (i.e., hidden parameter) of the environments themselves. A "probing" policy is run for a specified number of time steps. From the trajectory generated, a VAE is used to estimate the hidden parameter governing the transition dynamics. For the remaining time in the environment instance, a master policy, conditioned on the estimated parameter, is trained on the reward (any usual RL algorithm suffices; DDQN is used in the experiments).
The novelties of the approach are the problem setting (no access to optimal policies, immediate deployment at test-time), the joint training of a probing policy and VAE model, and a universal policy conditioned on the estimated hidden parameters. The experiments were conducted on domains that were, individually, both continuous and discontinuous with respect to the hidden parameters. The paper concludes that the proposed method improves upon baselines in terms of speed of reward attainment and computation time. Ablations were run for some of the components of the proposed algorithm.

Decision: Weak Reject

The two key reasons for the Weak Reject were the following.
1. Statistically insignificant results (3 runs)
2. Questions about baselines used (see question 6 below)

The proposed algorithm is promising and attempts to directly address limitations in previous literature. The paper thoroughly contextualizes and motivates the current approach in light of previous work in the literature and discusses possible advantages: simplicity and generality; the ability to deploy immediately at test-time instead of having to train another policy (from learning a universal policy); assuming that the hidden parameters are unobservable, which is more realistic; not assuming access to optimal value functions/policies; the algorithm is model-free, so one can avoid the computational and memory overhead (when compared to BNN) of learning a model and avoid compounding model error. I also appreciated the thorough discussion of the key algorithmic choices and their trade-offs in section 3. The presentation of ideas is clear throughout.

However, weak experiments keep this paper at a weak reject. The paper claims that the proposed algorithm "significantly outperforms" the baselines (abstract); however, more runs are needed to substantiate these claims. Only 3 runs were made for the test episode, which is not enough to be able to make empirical claims with non-trivial confidence (see https://arxiv.org/abs/1709.06560).  As well, since baseline methods are adapted from the original papers, more thorough hyperparameter sweeps should be performed. I wasn’t sure what “coarse coordinate search” was (pg. 6), but in general it would be good to write down the hyperparameters swept over.

Some additional questions about the experiments:
1. Did you try oracle with lower dimensional hidden parameter embedding? (understanding more why oracle does possibly worse in HIV and Acrobot)
2. There is the claim on page 2 that the proposed algorithm is "orders of magnitude faster than best model-based method". What is the best model-free method?
3. Is the universal policy of SEPT optimized during test time?
4. Did you manage to replicate the results of DPT in Yao (2018)? Were architectures the same?
5. What is the significance of not requiring rewards at test-time?
6. Why were BNN, MAML, and EPOpt the baselines chosen? What about the other works mentioned in the related work section, like Tirinzoni (2018) or Paul (2019)? Did you try using an LSTM as a baseline to directly learn the policy? One could imagine treating the existence of a hidden parameter as inducing a partial observability problem.
7. What is a "coarse coordinate search" over hyperparameters? What hyperparameters were tried?
8. Did you try regular DQN? DDQN w/o prioritized replay?
9. Why were the cumulative reward curves not as significant for Acrobot and HIV as compared to 2D navigation?
10. Why different probe length settings for 2D navigation vs. Acrobot and HIV? How did these hyperparameters affect the time to solve?
11. Why do 2D-room and Acrobot have terminal rewards of 1000 and 10?

I would be willing to raise the score if the key experimental problems I noted were addressed.

Minor comments/questions that did not impact the score
1. Should include the table of computational time in the main body if possible since computational efficiency is one of the core claims
2. Move DDQN comment in 3.2 to the experiments section
3. How does the probing policy approach relate to work in active perception? (https://link.springer.com/article/10.1007/s10514-017-9666-5)


EDIT: After the comments below which addressed my concerns, I have raised the score for this paper from a weak reject to an accept.



**Experience Assessment:**

I have read many papers in this area.

**Review Assessment: Checking Correctness Of Derivations And Theory:**

I carefully checked the derivations and theory.

**Review Assessment: Checking Correctness Of Experiments:**

I carefully checked the experiments.

**Review Assessment: Thoroughness In Paper Reading:**

I read the paper thoroughly.

---

> ### Author Response · Authors · 2019-11-15
> **[Part 1] We sincerely thank Reviewer 4 for a detailed and constructive review. We have scaled up to 20 independent training runs and main conclusions are unchanged.**
>
> We thank Reviewer 4 for recognizing the novelties of both the problem setting and algorithmic contributions of our work. We also appreciate the detailed review and constructive questions.
>
> We agree with Reviewer 4 on the benefit of conducting more independent training runs. We have scaled up our pipeline and updated the results in Figure 2. We re-trained all models in the paper for all 3 environments with 20 training runs (i.e., 20 random seeds) for each of SEPT, BNN, MAML, EPOpt-adv, Average, Oracle, MaxEnt, SEPT-NP, and TotalVar, with two minor exceptions. The main conclusions remain unchanged: SEPT can perform single episode transfer learning effectively, performing better than all baselines when z has discontinuous effect on the environment’s dynamics (2D navigation), better than all baselines when dynamics are continuous in z in HIV, while being robust in the case of Acrobot where most baselines except BNN perform well.
> (The two exceptions are: 1. we have not finished 20 runs of BNN on HIV due to its significantly longer computation time, but the original results showed BNN has negligible rewards during test; 2 we had insufficient time for SEPT-NP and DynaSEPT on HIV but will complete them.)
>
> >  Why were BNN, MAML, and EPOpt the baselines chosen? What about the other works mentioned in the related work section, like Tirinzoni (2018) or Paul (2019)?
>
> - There are very few appropriate baselines to compare against for single episode policy transfer. This is because most prior methods were not designed with the two constraints in mind: 1) the policy is deployed and evaluated on a single test episode with different unknown dynamics, and 2) no rewards are available at test time. Our main motivation is to transfer a policy trained on, for example, a simulation of a human patient to an actual human patient. In this case, we only get one shot at the test instance (i.e., actual patient), and we likely will receive no rewards at test. The only method we are aware of that has the same aim, and is therefore a “fair” comparison, is Direct Policy Transfer (DPT, [3]).  However, the authors of DPT have not made their code available. Nevertheless, we outperform their published results on the same environments (see Section 6).
> - Due to the lack of methods to compare against, we include comparisons to an Oracle policy, which receives the true hidden parameter values, and an “Average” policy, where the policy trained on training instances is run directly on the test instance.
> - We include MAML, EPOpt, and BNN as additional baselines to provide additional context to our results, even though those approaches were not specifically designed to perform well in the context of single episode transfer. MAML requires rewards at test time, while both MAML and BNN may need multiple episodes to adapt (for MAML) or to fine-tune the model (for BNN).
> - We have not included Tirinzoni (2018) since we believe MAML is representative of that approach, is state-of-the-art, and outperforms Tirinzoni in many cases. We have not included Paul 2019 since it requires the “distribution over hidden parameters to be known or controllable.” Our work considers the case where it is not known or controllable, which is true for precision medicine applications, among others.
>
> > Did you try using an LSTM as a baseline to directly learn the policy? One could imagine treating the existence of a hidden parameter as inducing a partial observability problem
>
> - As noted in [1], each set of task instances can be viewed as a special case of POMDPs in which 1) hidden parameters are constant during an episode and 2) given hidden parameters, the environment is Markovian. Once an episode begins, it is an MDP, and thus observing a history of states does not provide any additional information. Without history dependence, an LSTM (or frame concatenation, for example) provides no benefit over an MLP. Therefore we (and other HiP-MDP works cited [1,3,4]) do not include an LSTM-based policy as a baseline.
> - Consider 2D Nav, for example. No amount of history (sequences of states) provides any information regarding whether controls have flipped, and thus an LSTM would not be able to use that information to select better actions than an MLP policy.
>
> > Only 3 runs were made for the test episode
> To be clear, our results (for each method and for each domain) are the outcome of 20x (previously 3x) independent test episodes, where x is 10 for 2D navigation, 5 for Acrobot, and 5 for HIV. As we explain in Section 5, we conduct 20 (previously 3) independent training runs, and the resulting models are tested on multiple independent test instances (i.e., episodes).

---

> > ### Author Response · Authors · 2019-11-15
> > **[Part 2] Our responses to additional questions about the experiments**
> >
> > Reply to additional questions:
> > 1 Our new results with 20 training runs per method per environment with different random seeds show that the percentage difference between Oracle and SEPT is reduced. A lower-dimension embedding may be meaningful to try given a sufficiently large number of unique ground truth latent variables, but there may be difficulties for the case of HIV where dynamics were found to be unstable for certain settings [4].
> > 2 To the best of our knowledge, there is no prior experimental work that applied model-free RL to the single episode transfer problem in the 2D navigation, Acrobot, and HIV domains.
> > 3 The universal policy is not optimized during test time. It is only run in forward mode to generate actions given state and the estimated z.
> > 4 We were unable to find sufficient details in Yao et al. 2018 to replicate their experiments. However, assuming that they use the same BNN and DDQN as Killian et al. 2017, we outperform them on 2D navigation and Acrobot, while it is not comparable on HIV as they use different ground truth parameters from Killian 2017. The values were not reported, but their results suggest they were different from those used in Killian 2017.
> > 5 Not requiring access to rewards at test time means SEPT has greater potential applicability in real world settings. Taking precision medicine for example, training may be conducted in a simulation where the reward function depends on physiological factors that cannot be measured in practice. In fact, this is a very powerful feature because what we can measure on human patients is extremely limited relative to what we can “measure” in simulated patients.  Not requiring reward at test time means it is possible to train SEPT on simulated patients and deploy it on real test patients.
> > 6 See more detailed response above
> > 7 Coordinate search refers to https://en.wikipedia.org/wiki/Coordinate_descent, which is common practice when computational resources do not allow a full gridsearch, e.g. as used in Hessel et al [2]. We included the starting value of epsilon for DDQN, the dimension of latent variable, and length of the probe phase for SEPT.
> > 8 While virtually any RL algorithm can be plugged into SEPT (and many of the baselines), the focus of this work was to investigate the performance of transfer learning, given a fixed RL algorithm (DDQN) for all methods.
> > 9 As we explain in Section 6, differences between SEPT and baselines are much more significant in 2D navigation than in Acrobot and HIV, because dynamics is a continuous function of hidden parameters in Acrobot and HIV but discontinuous in 2D navigation. In 2D navigation, the effect of an action is flipped. Interpolation within policy space, which is done by the baselines (Average, EPOpt-adv and MAML), gives a feasible policy in the former case but makes much less sense for those methods in the latter case.
> > 10 2D navigation only requires 2 steps for the probe length because 2 steps are sufficient to distinguish the discontinuous dynamics (whether the same action causes movement left or right). HIV has more complex dynamics that are continuous in the hidden parameter, so requires a longer probe length. We acknowledge that the probe length is a hyper-parameter that should be chosen in accordance with the environment.
> > 11 The terminal reward of 2D navigation and Acrobot were chosen in Killian et al. 2017, which we follow for a fair comparison.
> >
> > Regarding hyperparameters:
> > We used almost exactly the same hyperparameters for DDQN as Killian et al 2017 [4] (i.e., the method we label as ‘BNN’), since we use the same three experiment domains as their work. The only hyperparameter that differed for DDQN is the starting value of epsilon in HIV, where we found 0.3 worked best for all methods. Our adaptation of EPOpt-adv only introduces an additional scalar threshold parameter, which was set to 10% in all experiments in Rajeswaran et al. 2017 [5]. As MAML does not specify when to take the adaptation step in the single episode setting, we choose it to be equal to the probe length of SEPT, so that we ensure MAML has equal number of remaining time steps for control. To the best of our knowledge, we are the first to report the performance of the simple baselines called Average and Oracle in this single episode problem setting. All methods use the same implementation of DDQN for the control policy.

---

> > > ### Author Response · Authors · 2019-11-15
> > > **[Part 3] Minor questions and references**
> > >
> > > Reply to minor comments:
> > > 1 Yes, we will consider including the tables of wall-clock train and test times in the main text. The tables show that all methods are on the same order of magnitude except for BNN, so our claim on runtime was only limited to comparing against BNN, which is the state-of-the-art model-based method on these tasks.
> > > 2 This makes sense. We can do that.
> > > 3 Satsangi et al. 2018 focus on modeling and planning in POMDPs, while we take a model-free approach. As explained in Konidaris et al. 2014 [1], the class of problems in our work can be viewed as a special case of POMDPs, since the latent variable that determines dynamics may be viewed as the “true unobserved state” and is assumed to be constant during an episode.
> > >
> > > [1] Konidaris and Doshi-Velez. Hidden Parameter Markov Decision Processes: An Emerging Paradigm for Modeling Families of Related Tasks. AAAI 2014.
> > > [2] Hessel et al. Rainbow: Combining improvements in deep reinforcement learning. AAAI 2018.
> > > [3] Yao et al. Direct Policy Transfer via Hidden Parameter Markov Decision Processes. LLARLA Workshop, FAIM. 2018
> > > [4] Killian et al. Robust and efficient transfer learning with hidden parameter markov decision processes. NIPS 2017
> > > [5] Rajeswaran et al. Epopt: Learning robust neural network policies using model ensembles. ICLR 2017.

---

> > > > ### Comment · AnonReviewer4 · 2019-11-15
> > > > **Response**
> > > >
> > > > Thanks for the detailed response! I will raise my score accordingly.

---

### Official Review · AnonReviewer2 · 2019-10-29
**Official Blind Review #2**

**Rating:** 3

**Review:**

This paper presents a strategy for single-trajectory transfer of a reinforcement learned policy.  They follow a typical approach in the few-shot supervised-learning community, of assuming that the plausible set of solutions may be modelled as a much lower dimensional latent variable, and then try to quickly infer that latent variable at test time.  In this case, the latent variable is ‘Z’.
At test time, they first run an exploitative algorithm (they call this the probe) to rapidly infer the value of Z, and thereby the optimal policy which this indexes, before switching to running the policy alone.

Weak reject.

I think that this paper (if indeed novel) is interesting, and I do agree that few-trajectory transfer in RL is a potentially impactful area in which to be working.  I think that the magnitude of the contribution is medium->low, however (namely that they have an approach which is wall-clock efficient, for single trajectory transfer), however the precise details of this contribution/claim have not been adequately tested:
We do not see experiments which highlight the wall-clock or inference cost competitiveness of the approach, and we do not see experiments against other, existing, approaches for latent-variable based RL (mostly these involve model-based RL, however these would probably be easy to derive for many of the test cases).
We also don’t see any experiments which evaluate the performance of SEPT as the time spent optimising the probe policy is varied.
As an aside, in figure 2(e) it appears that the Oracle policy is significantly outperformed by SEPT, do you have any thoughts as to why this might be?

To improve the paper, I would propose a slight rewrite to focus only on the core claim (i.e. why this model outperforms anything else for wall clock competitive single episode RL policy transfer).  I’d also be interested to see how this approach scales as the size of the potential policy set increases (e.g. if we had to produce a policy which would generalise for for all ball sports).
Any theoretical work would also be very welcome—one may be able to draw on existing work in the k-shot community to make a judgement as to what fraction of the trajectory is required before one should have a low entropy estimate of Z, and it would also be nice to know how the approach scales as the complexity of Z ramps up—i.e. as we need to support more tasks.

**Experience Assessment:**

I have read many papers in this area.

**Review Assessment: Checking Correctness Of Derivations And Theory:**

I assessed the sensibility of the derivations and theory.

**Review Assessment: Checking Correctness Of Experiments:**

I assessed the sensibility of the experiments.

**Review Assessment: Thoroughness In Paper Reading:**

I read the paper at least twice and used my best judgement in assessing the paper.

---

> ### Author Response · Authors · 2019-11-15
> **We appreciate the detailed questions and clarify our main contributions**
>
> We thank Reviewer 2 for detailed questions.
>
> > I think that this paper (if indeed novel) is interesting, and I do agree that few-trajectory transfer in RL is a potentially impactful area in which to be working.  I think that the magnitude of the contribution is medium->low, however (namely that they have an approach which is wall-clock efficient, for single trajectory transfer)
>
> Although wall-clock time at test time must be sufficiently low for real-time decision making problems, wall clock efficiency is not the main contribution of our paper. In Section 1, we mentioned computation time during test only in comparison to the state-of-the-art model-based method (which we call BNN). In fact, as Appendix D.2 shows, most baselines except BNN are comparable for both train and test. All methods except BNN are very fast at test time - sufficiently fast for real-time decision making. Instead, our main contribution (stated in Section 1) is a method for transfer learning for different unknown dynamics in RL, with a new challenging constraint where 1) we have only one “shot” with the test instance and 2) we have no rewards at test time. One of our motivating applications is the problem of transferring a policy trained on a simulation of a human patient to an actual patient. The “environment” dynamics of the real patient will be different from, but similar to, the simulated patient because there are properties of patients that affect dynamics which cannot be measured or observed.
>
> Consequently, we do not believe that wall clock time is the key metric.  Instead we consider cumulative reward to be the key metric. That said, the fact that SEPT is fast at test time is a secondary benefit, and is a result of the fact that it only requires running the learned models in forward mode, without any backprop or additional learning at test time.
>
> Regarding novelty: we are aware of only one existing method that addresses the need for policy transfer within a single episode without rewards at test time as described above: Direct Policy Transfer (DPT, [1]) The authors of DPT have not made their code available. Nevertheless, we outperform their reported results on the same environments (see Section 6).
>
> Regarding the probe:
> To avoid any potential misunderstanding, this probe is a policy. It is trained using simple policy gradient on the training instances, but deployed without additional learning on the test instance.
>
> Regarding “time spent optimising the probe policy”:
> We believe that the wall-clock time allowed for optimizing a policy is a less relevant measurement of computational efficiency in the RL literature. The practice in the literature is to report number of steps in the environment. Hence, the meaningful hyperparameter to vary for the probe policy is the number of environment time steps used during the probe phase of each episode, which we show in Figure 3(a-c). The wall-clock time used to train the probe policy is part of the training time, which we report in Table 2 in Appendix D.2.
>
> Regarding Figure 2e (“Oracle policy is significantly outperformed by SEPT”):
> We have updated Figure 2 with 20 independent runs for all methods, and specifically in Figure 2e we see that the percentage difference between Oracle and SEPT is reduced (although still significant). As stated in the main text in Section 6, the Oracle’s low performance in HIV may be due to the high-dimensionality of the ground truth hidden parameters. The ground truth has dimension 12, in contrast to the reduce latent dimension of 6 used by SEPT. Similar to Killian et al. 2017, we used five training instances for HIV, which may be insufficient for the Oracle to generalize in 12 dimensions.
>
> Regarding theoretical analysis:
> We agree that the theory of transfer learning in RL, especially with a single episode test constraint, is worth more effort from the community. We focused on contributing a practical algorithm with extensive ablations and comparisons to baselines, to show that this research direction can deliver promising experimental results. This can complement efforts on the theoretical front and suggest directions for theoretical work.
>
> [1] Yao et al. Direct Policy Transfer via Hidden Parameter Markov Decision Processes. LLARLA Workshop, FAIM. 2018

---

### Public Comment · ~Roberta_Raileanu1 · 2019-11-07
**Comment on the Acrobot Environment Setup**

Thank you for an interesting submission and for your work towards more realistic settings.

I wanted to bring something to your attention. I was curious how other methods would perform on the proposed environment setup, so I ran some experiments on Acrobot, using the same training and validation dynamics as the ones specified in the paper. While experimenting with this environment, I’ve found that PPO policies trained on the training environments perform quite well on the validation ones, without any extra fine-tuning. For example, it seems that a PPO policy trained on dynamics described by dz = 0.3 (part of the training set used in the paper) solves the validation tasks in 85 steps on average, which is better than the performance of all baselines presented and within the standard deviation of SEPT’s performance (79 +/ 23).

Since a model trained on a single environment performs so well, perhaps this environment setup is not very well suited for testing adaptation to diverse and unseen dynamics. It might be useful to include such simple baselines to show that the environments require significant adaptation.

---

> ### Author Response · Authors · 2019-11-15
> **We clarify the importance of measuring the difficulty of transfer learning with respect to a given RL algorithm**
>
> We appreciate your interest in this work. We agree that it is worthwhile to try different RL algorithms besides DDQN, since using a stronger algorithm for the RL component of all the methods can in principle improve the performance for all methods. However, the fact that PPO meets or exceeds the performance of methods that use DDQN for the RL component should not be used to conclude that the task itself is not well suited for testing the ability to handle unseen dynamics. The difficulty of a transfer or generalization problem must be measured with respect to a fixed reference point, which in this case is the RL algorithm that is plugged in to all methods. This is the reason that previous work concluded that Acrobot poses a challenge for transfer and generalization [1,2]; the fact that they made this finding due to their use of simpler and less powerful RL algorithms does not not invalidate their conclusion regarding the task. Given a fixed RL algorithm, the performance gap between the methods that we call Oracle and Average show the potential improvement to be made by accounting for latent variables that underlie differences in dynamics. However, even the Oracle faces a difficult function approximation problem when there are too few ground truth latent variable values for it to generalize. To get a better sense of the performance gap, one can compare with a policy that is only trained on the test instance.
>
> [1] Bai et al. Planning how to learn. ICRA 2013.
> [2] Killian et al. Robust and Efficient Transfer Learning with Hidden Parameter Markov Decision Processes. NIPS 2017

---

### Decision · Program_Chairs · 2019-12-19

**Decision:**

Accept (Poster)

**Comment:**

This is an interesting paper that is concerned with single episode transfer to reinforcement learning problems with different dynamics models, assuming they are parameterised by a latent variable. Given some initial training tasks to learn about this parameter, and a new test task, they present an algorithm to probe and estimate the latent variable on the test task, whereafter the inferred latent variable  is used as input to a control policy.

There were several issues raised by the reviewers. Firstly, there were questions with the number of runs and the baseline implementations, which were all addressed in the rebuttals. Then, there were questions around the novelty and the main contribution being wall-clock time. These issues were also adequately addressed.

In light of this, I recommend acceptance of this paper.